# Th2 Cells Are Associated with Tumor Recurrence Following Radiation

**DOI:** 10.3390/cancers16081586

**Published:** 2024-04-20

**Authors:** Mohamed K. Abdelhakiem, Riyue Bao, Phillip M. Pifer, David Molkentine, Jessica Molkentine, Andrew Hefner, Beth Beadle, John V. Heymach, Jason J. Luke, Robert L. Ferris, Curtis R. Pickering, Jing H. Wang, Ravi B. Patel, Heath D. Skinner

**Affiliations:** 1Department of Radiation Oncology, UPMC Hillman Cancer Center, University of Pittsburgh, Pittsburgh, PA 15232, USA; abdelhakiemm@upmc.edu (M.K.A.); ppifer@hsc.wvu.edu (P.M.P.); hefnera@upmc.edu (A.H.); patelr20@upmc.edu (R.B.P.); 2Department of Medicine, UPMC Hillman Cancer Center, University of Pittsburgh, Pittsburgh, PA 15232, USA; baor@upmc.edu (R.B.); lukejj@upmc.edu (J.J.L.); wangj28@upmc.edu (J.H.W.); 3Department of Thoracic-Head and Neck Medical Oncology, The University of Texas MD Anderson Cancer Center, Houston, TX 77030, USA; dmolkentine@mdanderson.org (D.M.); jmolkentine@mdanderson.org (J.M.); jheymach@mdanderson.org (J.V.H.); 4Department of Radiation Oncology, Stanford University, Stanford, CA 94305, USA; bbeadle@stanford.edu; 5Department of Otolaryngology, UPMC Hillman Cancer Center, University of Pittsburgh, Pittsburgh, PA 15232, USA; ferrrl@upmc.edu; 6Department of Surgery—Otolaryngology, Yale University, New Haven, CT 06520, USA; curtis.pickering@yale.edu

**Keywords:** tumor microenvironment, immune microenvironment, biomarkers, radiation, resistance

## Abstract

**Simple Summary:**

Locoregional recurrence following the definitive treatment of head and neck cancers often leads to patient demise. Radiation is frequently used in the definitive treatment of these cancers. Resistance to radiation leads to increased locoregional recurrence, particularly in patients with human papillomavirus-negative head and neck squamous cell carcinomas. In this study, we show that tumor Th2 immune infiltrate is associated with increased locoregional recurrence in patients with human papillomavirus-negative head and neck squamous cell carcinoma. These findings can be used to develop novel immunotherapeutic approaches to improve the response to radiation and thus decrease locoregional recurrence and improve outcomes in patients with human papillomavirus-negative head and neck squamous cell carcinomas.

**Abstract:**

The curative treatment of multiple solid tumors, including head and neck squamous cell carcinoma (HNSCC), utilizes radiation. The outcomes for HPV/p16-negative HNSCC are significantly worse than HPV/p16-positive tumors, with increased radiation resistance leading to worse locoregional recurrence (LRR) and ultimately death. This study analyzed the relationship between immune function and outcomes following radiation in HPV/p16-negative tumors to identify mechanisms of radiation resistance and prognostic immune biomarkers. A discovery cohort of 94 patients with HNSCC treated uniformly with surgery and adjuvant radiation and a validation cohort of 97 similarly treated patients were utilized. Tumor immune infiltrates were derived from RNAseq gene expression. The immune cell types significantly associated with outcomes in the discovery cohort were examined in the independent validation cohort. A positive association between high Th2 infiltration and LRR was identified in the discovery cohort and validated in the validation cohort. Tumor mutations in CREBBP/EP300 and CASP8 were significantly associated with Th2 infiltration. A pathway analysis of genes correlated with Th2 cells revealed the potential repression of the antitumor immune response and the activation of BRCA1-associated DNA damage repair in multiple cohorts. The Th2 infiltrates were enriched in the HPV/p16-negative HNSCC tumors and associated with LRR and mutations in CASP8, CREBBP/EP300, and pathways previously shown to impact the response to radiation.

## 1. Introduction

The curative treatment of nearly all locally advanced solid tumors involves radiation; however, there are few biologically derived drivers of response to radiation. Moreover, although the immune response impacts tumor radioresistance in preclinical models [1], the clinical impact of specific tumor-infiltrating lymphocytes (TILs) on the radiation response is unclear.

Head and neck squamous cell carcinoma (HNSCC) leads to the death of over 360,000 patients annually worldwide [2]. Outcomes are driven by tumor characteristics and human papillomavirus (HPV) status, with HPV-positive tumors exhibiting better outcomes [3]. In HNSCC, locoregional recurrence (LRR) following radiation, and not distant metastasis (DM), is the most common cause of cancer death [4]. Thus, radioresistance is a direct cause of death in HNSCC, particularly in HPV-negative disease. While there are no clinically utilized biomarkers of therapeutic resistance for HPV-negative tumors, genes such as TP53, CREBBP, EP300, and CASP8 may predict radioresistance [5,6,7,8].

Radiation relies on immune activation to exert maximal antitumor effects [9] via multiple pathways, including the release of tumor antigens and proinflammatory cytokines/chemokines and the activation of TILs and dendritic cells [10,11]. Furthermore, PD-L1 overexpression has been associated with worse LRR following radiation [12]. These observations, coupled with anti-PD1 activity in HNSCC [13,14], has led to clinical trials of concurrent immunotherapy and radiation [15,16,17]. Unfortunately, these studies did not meet their primary endpoints, although post hoc analyses have indicated some benefit in PD-L1-high patients [16,17,18,19]. Regardless, these results highlight the need for additional study of the intersection between curative radiation and the immune response.

In this study, we utilized two HPV-negative HNSCC populations treated with surgical resection and adjuvant radiation to examine TILs associated with LRR. Pretreatment tumors from both cohorts were digitally dissected for the abundance of TILs using an integration of gene set enrichment analysis and deconvolution approaches (xCell) [20]. Univariable and multivariable analyses for the association between immune cell type and clinical outcome were performed in the discovery cohort. Significant cell types from this analysis were then examined in the independent validation cohort. High-T helper 2 (Th2) infiltrate was identified as being significantly associated with LRR in both cohorts. Additional analysis was performed to identify mutations and pathways reliably associated with Th2 cell infiltration.

## 2. Materials and Methods

### 2.1. Discovery Cohort and Validation Cohort Clinical Outcome Datasets

The discovery cohort reflects 94 patients from the TCGA cohort with HPV-negative HNSCC treated with surgery and postoperative radiation, with detailed treatment and clinical outcomes [7]. The validation cohort consisted of 97 HPV-negative HNSCC patients treated with surgery and postoperative radiation for which expression data necessary for TIL prediction were available [12]. Tumors were staged using AJCC 7th edition.

### 2.2. Genomic and Transcriptomic Datasets

Tumor mutation and RNA-seq data available for the discovery cohort and the TCGA cohort have been previously described and downloaded from cBioPortal [21], while Illumina mRNA array data from the validation cohort have been published previously [22]. Unique (e.g., nonoverlapping with TCGA) RNA sequencing data from the International Cancer Genome Consortium (ICGC) data portal [23,24] (accessed on 1 October 2021) were used to validate the pathway analysis. A separate pathway validation was performed using additional RNA Seq data from the NCI’s Clinical Proteomic Tumor Analysis Consortium (CPTAC) HNSCC cohort (accessed on 1 November 2021) [25].

### 2.3. Digital Dissection of the Tumor Microenvironment for TIL Abundance

RNAseq data from both the discovery and validation cohorts were used to predict TIL abundance in tumors using xCell [20] (https://xcell.ucsf.edu/) (analyzed on 10 September 2021). xCell quantifies 64 immune, stromal, and tumor cell abundances in a heterogeneous tumor microenvironment from tissue RNA-seq data [20]. Specifically, -xCell was used to digitally deconvolute cell populations from RNAseq data using R library xcell:: xCellAnalysis with default settings. This method employs both ssGESA and deconvolution algorithms to compute the scores for each cell population in a sample, including Th2 cells, among others. All samples were organized based on their Th2 cell scores, which were generated using xCell. Subsequently, they were divided into three groups—T1, T2, and T3—with each group representing one-third of the total number of immune cell types present in at least 15 tumors and significantly associated with clinical outcome in the discovery cohort were selected for further validation using the validation cohort.

### 2.4. Identification of Somatic Mutations Associated with Th2 Cell Infiltrates

Somatic mutations deemed significant in the TCGA cohort according to MutSig [26] and identified as cancer genes in OncoKB [27] were determined. Average derived Th2 infiltrate was compared between mutant versus wild type for each mutation using one-way ANOVA and post hoc comparison (SPSS v27). Unadjusted *p*-values are shown for survival analysis. For pathway analysis, the multiple comparisons effect was adjusted using the Benjamini–Hochberg (BH) method [28].

### 2.5. Identification of Genes Positively or Negatively Correlated with Th2 Cell Infiltrates by Expression

Ingenuity pathway analysis (IPA) (Qiagen) was performed to determine enriched pathways and activated z-scores for genes with ≥1.5-fold difference between the upper and lower tertiles of derived Th2 infiltrate at an FDR of 0.05 in the HPV-negative TCGA cohort (*n* = 409). Significance of pathway enrichment was determined using the entire human genome as the reference set. The analysis of the IPA was performed with species selected as “Human, Mouse, Rat” and other parameters as default. The reference set in the IPA was set to “Ingenuity Knowledge Base (Genes only)” with analysis performed on 9 November 2021.

### 2.6. Survival Analysis

Overall survival (OS) was defined as the time from diagnosis until death or the last follow-up. Time to LRR or DM was defined as the time from diagnosis until either an event or the last follow-up. Clinical variables included site, nodal stage (0–1 versus ≥ 2a), and tumor stage (1–2 versus 3–4). xCell z-scores were transformed by log10 (score × 100 + 1) for the survival analysis. Univariable Cox proportional hazards (PH) model was used to test the associations between the abundance of each immune infiltrate and outcomes. Variables significant on univariable analysis were included in multivariable analysis where indicated. Kaplan–Meier analysis (SPSS (v27)) was performed using a log-rank test comparing the lower versus upper two tertiles of derived Th2 cells in both cohorts.

## 3. Results

The tumor characteristics of the discovery cohort are shown in Table 1. All the patients had HPV/p16-negative tumors and were treated with ~60 Gy following resection. Most of the patients had tumors of the oral cavity (63.8%) or larynx/hypopharynx (31.9%), with a small number of oropharyngeal tumors (4.3%). Most of the tumors were staged as either T3 (24.5%) or T4 (59.6%), with 48.9% of the patients found to exhibit nodal involvement following resection.

### 3.1. Th2 Cell Type Is Significantly Associated with Clinical Outcome in Two Similarly Treated HNSCC Cohorts

We examined the discovery cohort for an association between the clinical variables and cell types with OS. The nodal stage was significantly associated with OS (*p* = 0.03). The presence of monocytes (HR 2.1, 95% CI 1.008–4.34, *p* = 0.05) was not significantly associated with OS; however, both Th2 cells (HR 6.1, 95% CI 1.7–21.7, *p* = 0.005) and CD4+ memory T cells (HR 5.9, 95% CI 1.7–20.6, *p* = 0.006) were significantly associated with a worse OS (Figure 1A). On the multivariable Cox regression analysis, the nodal stage (*p* = 0.008), monocytes (*p* = 0.02), and Th2 cells (*p* = 0.008) remained significant.

We next evaluated the association between the intratumoral cell populations and DM or LRR. The time to DM was significantly associated with a higher nodal stage (*p* = 6.4 × 10^−4^). In addition, the CD4+ memory T cells (HR 11.63, 95% CI 1.8–75.9, *p* = 0.01), sebocytes (HR 0.068, 95% CI 0.01–0.48, *p* = 0.007), and epithelial cells (HR 8.7 × 10^−3^, 95% CI 3.56 × 10^−4^–0.21, *p* = 0.01) were significantly associated with DM on the univariable analysis (Figure 1B). After adjusting for the nodal status in a multivariable model, the presence of CD4+ memory cells remained significantly associated with DM (*p* = 0.012). We next examined LRR and found no significant association with any clinical variables evaluated. However, LRR was significantly associated with specific TILs, including common lymphoid progenitor cells (CLPs) (HR 9.5 95% CI 1.21–74.9, *p* = 0.03), Th2 cells (HR 9.0 1.4–57.0, *p* = 0.02), CD4+ memory T cells (HR 7.351843, 95% CI 1.3–41.3, *p* = 0.024), and immature dendritic cells (IDCs) (HR 0.34, 95% CI 0.17–0.67, *p* = 0.002) (Figure 1C).

To validate the associations seen between the intratumoral cell types and clinical outcome, we used gene expression data from an independent patient population treated in a similar fashion [12]. We repeated the same analysis using xCell to predict the cell type abundance in this validation cohort and focused only on the cell types identified as being significant in the discovery cohort. None of the cell types identified in the discovery cohort were found to be significant for OS or DM in the validation cohort. However, we found that the presence of Th2 cells was associated with significantly worse LRR both in the discovery and validation cohorts when the same cutoff value was used (lower tertile vs. others) (*p* = 0.019) (Figure 1D).

### 3.2. Somatic Mutations and Gene Expression Associated with Th2 Cell Infiltration

To explore the relationship between tumor signaling and Th2 infiltrate, we examined differences in the presence of Th2 based upon tumor mutation in the entirety of the HPV-negative TCGA cohort for which both mutational and gene expression data were available (*n* = 406). Mutations in several genes, specifically CASP8, HRAS, and HLA-A, were associated with an increased presence of derived Th2 infiltrate (Figure 2A). Although neither EP300 nor CREBBP mutations were individually associated with increased Th2 infiltrate, if taken together, mutations in these functionally similar histone acetyltransferases were associated with increased Th2 infiltrate (*p* = 0.042, Figure 2A). When correcting for multiple hypothesis testing, only CASP8 remained significant, with an FDR of 5%.

We next examined the relationship between the Th2 infiltrates and gene expression using the RNA-seq data of the HPV-negative TCGA cohort, for which only gene expression data were required (*n* = 409). These data were analyzed using IPA to identify pathways associated with the derived Th2 infiltrate. Several cancer- and immune-related pathways were identified (Figure 2B). For instance, the “BRCA1 in DRR” pathway and the “Tumor microenvironment (TME)” pathway were positively and negatively associated with the Th2 infiltrate, respectively. The signaling nodes and expression patterns for both pathways are shown in Figure A1 and Figure A2, respectively, with the nodes significantly positively or negatively correlated with the Th2 infiltrate.

To confirm the relationship between these two pathways and the derived Th2 infiltrate, pathway activation scores were generated as described in the Methods section. As expected, a high degree of positive correlation was observed between the derived Th2 infiltrate and BRCA1 in the DDR pathway (r = 0.73, *p* < 2.2 × 10^−16^), while a significant negative correlation was observed with the TME pathway (r = −0.45, *p* < 2.2 × 10^−16^) (Figure 3A). Pathway activation scores were generated using gene expression data from a separate nonoverlapping HPV-negative HNSCC cohort [22,23] and similar associations with the derived Th2 infiltrate were observed (Figure 3B). Specifically, the Th2 infiltrate was correlated with the activation of BRCA1 in the DDR pathway (r = 0.77, *p* = 5.6 × 10^−9^) and the repression of the TME pathway (r = −0.44, *p* = 0.007). We also analyzed RNA-Seq data from the separate HPV-negative CPTAC database and again found the same pathway associations, with the derived Th2 infiltrate associated with the activation of BRCA1 in DDR (r = 0.59, *p* < 2.2 × 10^−16^) and the repression of the TME pathway (r = −0.48, *p* = 1.6 × 10^−7^) (Figure 3C).

The correlation of the Th2 infiltrate with other pathways, specifically the Cell Cycle Control of Chromosomal Replication Pathway as well as the Kinetochore Metaphase Signaling Pathway was performed across the TCGA, ICGC, and CPTAC cohorts (Figure A3). In all the cohorts, both pathways are significantly positively correlated with the Th2 infiltrate.

## 4. Discussion

Patients with HNSCC are currently stratified by HPV/p16 status and clinical variables; however, there is a large variability in the response to therapy within these groups. Attempts to characterize HNSCC based on immune infiltrate and link these characterizations to outcomes have been hampered by small numbers, the heterogeneity of HNSCC itself, and a lack of uniformly treated patients [29,30,31].

Previously, our group demonstrated that high PD-L1 expression was associated with increased LRR after treatment with surgery and postoperative radiation in patients with HPV-negative HNSCC [12]. In contrast, patients with high CD8 TILs and low PD-L1 expression exhibited no LRR or death due to disease [12], suggesting that pretreatment TILs in the TME have a strong influence on outcomes. In this study, we build upon our previous work and investigate the TME in uniformly treated patients to elucidate additional biomarkers.

While the role of TILs in solid tumors is well known, the focus has primarily been on CD8+ or Th1 TILs as effector immune cells or regulatory T cells as suppressive immune cells. Conversely, Th2 cells and their signature cytokines (IL-4, IL-5, and IL-13) are largely characterized by their role in helminth infection or allergic responses and the promotion of IgE-mediated eosinophilic responses [32]. In the context of cancer, Th2 cells are considered to be protumorigenic [33], with potential protumorigenic actions of Th2 cells being reported, including the recruitment and repolarization of macrophages to suppressive (M2-like) lineages [34], the stimulation of the production of vascular endothelial growth factor (VEGF) [35], and the production of immunosuppressive cytokines [33]. The variability of the Th2 response in different preclinical tumor models may be in part due to plasticity between Th1 and Th2 polarization, with factors such as VEGF shown to repolarize T cells from a Th1 to a Th2 phenotype [36], while PD-1 blockade has been shown to revert the Th2 to Th1 phenotype [37]. The role of Th2 cells in the tumor microenvironment of clinical tumor samples is also unclear, with Th2 cells being associated with aggressive tumor histologies, including HNSCC [38], as well as histologies with better prognosis, such as Hodgkin’s lymphoma [39]. Analogous to our findings in this report, a recent study by Rui et al. demonstrated that high-Th2 infiltrate in prostate cancer was associated with increased recurrence after prostatectomy [40].

In this study, we also examined tumor genetic mutations associated with Th2 infiltrate. We found that mutations in HLA-A, CASP8, and HRAS correlated with the Th2 gene signature. Additionally, CREBBP/EP300 mutations were also associated with increased Th2 infiltrate. The CASP8 and CREBBP/EP300 mutations were of particular interest since we have shown that both mutations are associated with radioresistance in HNSCC [7,8]. However, caspase 8 has been shown to be critical in nuclear factor-κβ (NF-κβ) pathway activation, causing the IL-1 secretion required for the Th2 response [41]. This discordance between the dependence of Th2 function on caspase 8 and an association between CASP8 mutation and Th2 infiltrate may be at least partially explained by cancer-associated missense mutations in CASP8 associated with epithelial cancers that inhibit its apoptotic function but chronically activate NF-κβ signaling [42]. In patients whose tumors exhibit a Th2 gene signature, we found increased expression of genes related to the cell cycle and DNA repair, which can confer radioresistance. These effects may in part be generated through the Th2 cytokine IL-4, which has been shown to increase the expression of nonhomologous end joining DNA repair proteins and induce the proliferation of keratinocytes through c-myc-driven G0/G1-to-S phase cell cycle progression [43,44].

While our findings have revealed a novel association between outcomes and Th2 infiltrates in HPV-negative HNSCC, there are several limitations to our approach. Foremost, both our discovery and validation cohort analyses were performed in a retrospective fashion using prediction models for infiltrates based on mRNA expression data. Due to this limitation, further analyses of the TME in these samples, including direct TIL visualization, functional analyses of TIL infiltrates, or measurement of cytokine levels/activity, were not possible and will be performed in the future. Despite these limitations, our study did reveal a potential new prognostic biomarker for HPV-negative HNSCC that could be utilized to inform future treatment selection and clinical trial design.

The limitations of this study include the retrospective nature of the analyses, the limited sample size, and the inherent limitations of the transcriptomic, genomic, and proteomic analyses.

## 5. Conclusions

An increased Th2 immune infiltrate was associated with locoregional recurrence in a discovery cohort of HPV-negative head and neck squamous cell carcinoma patients treated with surgery and postoperative radiation. This was confirmed in a separate validation cohort of similarly treated HPV-negative head and neck squamous cell carcinoma patients. These findings point to potential immunotherapeutic approaches to augment the response to radiation and decrease LRR in HPV-negative HNSCC patients.

## Figures and Tables

**Figure 1 cancers-16-01586-f001:**
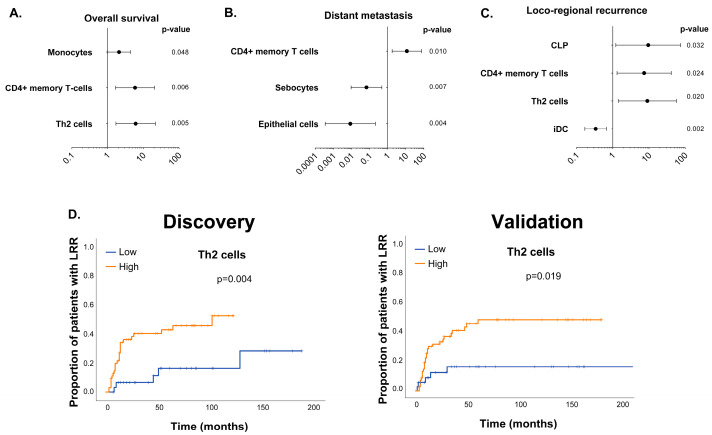
Association between immune subtype and outcome in HPV-negative HNSCC treated with radiation. (**A**–**C**) Forest plot of univariable analysis for overall survival (**A**), distant metastasis (**B**), and locoregional recurrence (**C**) following surgery and adjuvant radiation. Hazard ratios and 95% confidence intervals are shown for each clinical outcome listed. (**D**) Kaplan–Meier curve showing locoregional recurrence as a function of high (upper two tertiles) versus low (lower tertile) Th2 infiltrate in the discovery (*n* = 94) and validation cohorts (*n* = 97). The Cox PH model was used in (**A**–**C**). The log-rank test was used in (**D**).

**Figure 2 cancers-16-01586-f002:**
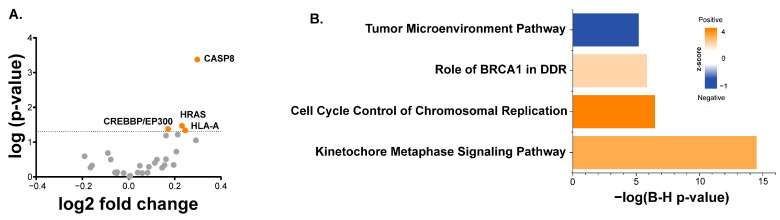
Tumor alterations associated with derived Th2 infiltrate. (**A**) Volcano plot of mutations in HNSCC and Th2 infiltrate from the full HPV-negative TCGA HNSCC cohort with mutational and gene expression data (*n* = 406). Mutations exhibiting significantly altered Th2 infiltrate (unadjusted *p*-value and fold change (mean mutant/mean wild type) are shown, with orange indicating that the mutation is associated with increased infiltrate). (**B**) Ingenuity pathway analysis of genes associated with Th2 from the full TCGA Head and Neck cohort (*n* = 409), including z-scores and B-H adjusted *p*-values.

**Figure 3 cancers-16-01586-f003:**
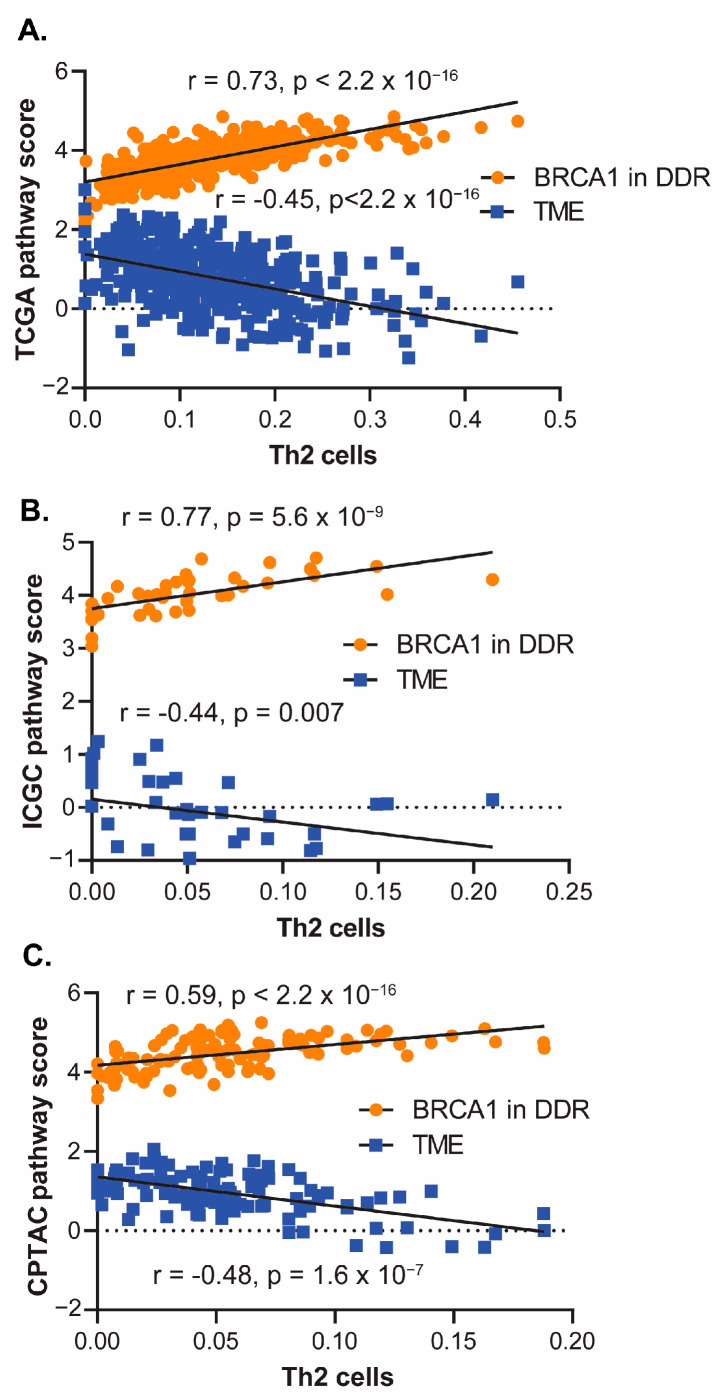
The DDR and TME scores are significantly associated with derived Th2 infiltrate in multiple clinical cohorts. BRCA1 in DDR and TME scores (as calculated in the Materials and Methods section) and derived Th2 infiltrate in the full HPV-negative TCGA cohort with available gene expression data (*n* = 409) (**A**), nonoverlapping HPV-negative HNSCC patients from the International Cancer Genome Consortium (ICGC) data (*n* = 40) (**B**), and the National Cancer Institute’s Clinical Proteomic Tumor Analysis Consortium (CPTAC) Head and Neck cohort (*n* = 110) (**C**). Correlation coefficient and *p*-value were computed by Spearman’s correlation in (**A**–**C**). *p*-value < 0.05 indicates the correlation coefficient is significantly different from zero.

**Table 1 cancers-16-01586-t001:** Tumor characteristics of the HPV-negative squamous cell carcinoma discovery cohort.

Nodal Stage	*n*	Percent
0	44	46.8
1	20	21.3
2x	1	1.1
2a	2	2.1
2b	13	13.8
2c	8	8.5
3	2	2.1
x	4	4.3
**Tumor stage**		
1	1	1.1
2	12	12.8
3	23	24.5
4	56	59.6
x	2	2.1
**Site**		
Oral cavity	60	63.8
OPX	4	4.3
Larynx/hypopharynx	30	31.9

## Data Availability

Data used in this publication were generated by The Cancer Genome Atlas Program (TCGA) (https://portal.gdc.cancer.gov/) (accessed on 1 October 2021), International Cancer Genome Consortium (ICGC) (https://dcc.icgc.org/repositories) (accessed on 1 October 2021), and Clinical Proteomic Tumor Analysis Consortium (CPTAC) (https://proteomic.datacommons.cancer.gov/pdc/) (accessed on 1 November 2021). Validation cohort data have been previously published [7].

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
