# Peer review of "Th2 Cells Are Associated with Tumor Recurrence Following Radiation"

_cancers, 2024, doi:10.3390/cancers16081586_

Round 1
Reviewer 1 Report (Previous Reviewer 1)
Comments and Suggestions for Authors
The rebuttal answers all my queries but some of information requested needs to be included in the manuscript.
Point 3: Thank you for the correlation plots of the other IPA-enriched pathways. Is there a rationale for not including this new data in Figure 3 or at least as supplemental to figure 3?
Point 4: Please include this information in the Methods section of the main text.
Point 7: Please include this information in the legend of Figure 3 (it doesn’t appear anywhere in the manuscript)
Author Response
Comments and Suggestions for Authors
The rebuttal answers all my queries but some of information requested needs to be included in the manuscript.
Point 3: Thank you for the correlation plots of the other IPA-enriched pathways. Is there a rationale for not including this new data in Figure 3 or at least as supplemental to figure 3?
- The plots were added as supplemental figures (Figure A3).
Point 4: Please include this information in the Methods section of the main text.
- This was included in the text in the Methods section as follows: "The analysis of IPA was performed with Species selected as “Human, Mouse, Rat” and other parameters as default. The Reference Set in IPA was set to “Ingenuity Knowledge Base (Genes only) with analysis performed on 11-09-2021."
Point 7: Please include this information in the legend of Figure 3 (it doesn’t appear anywhere in the manuscript)
- This was included in the legend of Figure 3: “Correlation coefficient and p-value were computed by Spearman’s correlation in A to C. p-value < 0.05 indicates the correlation coefficient is significantly different from zero.”
Reviewer 2 Report (Previous Reviewer 3)
Comments and Suggestions for Authors
My suggestions were included in revision.
Author Response
Thank you for taking the time to review our manuscript.
This manuscript is a resubmission of an earlier submission. The following is a list of the peer review reports and author responses from that submission.
Round 1
Reviewer 1 Report
Comments and Suggestions for Authors
Thank you for the clarifications and modifications that have addressed most of my original concerns. There are a few outstanding major concerns.
MAJOR
- Re figure 2A. These are unadjusted p-values. Please correct for multiple hypothesis testing. If these genes dropout, they should be removed from discussion and you can speak to trends in these cohorts and the robustness in previously published data. It might be robust in other cohorts, but doesn’t seem to be in these ones.
- “The predicted presence of Th2 cells is generated via xCell using the expression of specific genes…that are almost completely distinct from the genes that make up either pathway.” This is interesting. It would be informative to the reader to know the overlap of genes and the specific gene sets for each pathway. The identity of the genes associated with the pathways is minimum information for publications involving IPA (i.e., paywalled knowledge base) so that the information is accessible to all researchers. Please include this information (geneset enrichment data) in supplemental material
- As previously requested, please show the activity vs. Th2 infiltrate correlation data (even if r=0) for Cell cycle control and kinetochore metaphase pathway genesets across all 3 cohorts (as in Figure 3). This will add valuable information about the similarities and differences across these different cohorts.
MINOR
- Regarding background reference set used for IPA: The rebuttal mentions the reference set used for IPA was the entire human genome. This will introduce bias in enrichment scores given that only a subset of genes are expressed in the tumours. The minimum information needed for publishing reproducible IPA analysis includes this information. Please include this in the revised manuscript (I don’t see it included in the current revision).
- Lines 127-140: In your answer to the original query regarding this passage you indicate that this paragraph describes IPA’s algorithm for calculating activation scores. It is not needed in this context as this is already described in IPA literature and this renders the entire section confusing to the reader. Please remove and re-write similar to: “Ingenuity pathway analysis (IPA) (Qiagen) was performed to determine pathways enriched, along with activation z-scores for genes with 1.5-fold difference between the upper and lower tertiles of derived Th2 infiltrate at an FDR of 0.05 in the HPV-negative TCGA cohort (n=409). Significance of pathway enrichment was determined using the entire human genome as the reference set.”
- Reference 21. The title of this preprint with 3 different versions has changed 3 times. Please ensure the referenced title is for the latest version, or if the cited data is no longer in most recent version, please specifically site the version number of the preprint.
- Please include the information about use of Spearman’s corr and associated statistical test in Methods and/or Figure 3 legend.
- The legend inset of Figure 2B has 2 colour breaks (blue, orange) and no z-score values other than zero indicated, yet there are 4 different colours represented in the data. Please correct this.
Reviewer 2 Report
Comments and Suggestions for Authors
Manuscript cancers-2852669
„Th2 cells are associated with tumor recurrence following radiation” for Cancers
Comments:
1. Th2 cells are associated with the induction and development of humoral immune responses. I ask the authors to expand on this topic in the discussion and indicate what potential role activation of humoral immunity may play in the locoregional recurrence of head and neck squamous cell carcinomas.
Author Response
Thank you for the feedback, we are further examining our findings outlined in this manuscript and the role of Th2 cells in immune response and activation in ongoing and future projects to hopefully be able to better and more accurately describe this role in future manuscripts.
Reviewer 3 Report
Comments and Suggestions for Authors
This is a useful study that reports important data for further clinical approaches toward head and neck squamous cell carcinoma diagnosis and treatment.
1. Provide more information on Th2 cells (and full name when mentioned for the first time, the same in the title)
2. Add study limitation
3. Add a schematic drawing summarizing your results
